# Association Between Pre-Existing Conditions and COVID-19 Hospitalization, Intensive Care Services, and Mortality: A Cross-Sectional Analysis of an International Global Health Data Repository

**DOI:** 10.3390/pathogens14090917

**Published:** 2025-09-11

**Authors:** Basant M. S. Elsayed, Lina Altarawneh, Habib Hassan Farooqui, Muhammad Naseem Khan, Giridhara Rathnaiah Babu, Suhail A. R. Doi, Tawanda Chivese

**Affiliations:** 1Department of Population Medicine, College of Medicine, QU Health, Qatar University, Doha 2713, Qatar; belsayed@hamad.qa (B.M.S.E.); la1908280@student.qu.edu.qa (L.A.); hfarooqui@qu.edu.qa (H.H.F.); naseem@qu.edu.qa (M.N.K.); gbabu@qu.edu.qa (G.R.B.); sdoi@qu.edu.qa (S.A.R.D.); 2Department of Science and Mathematics, School of Interdisciplinary Arts and Sciences, University of Washington Tacoma, Tacoma, Washington, WA 98402, USA

**Keywords:** COVID-19, mortality, intensive care unit, hospitalization, pre-existing conditions, global health, data sharing

## Abstract

Background: The use of globally shared individual-level data in answering epidemiological questions during health emergencies of international concern is still debatable. In this study, we investigated the association between pre-existing conditions and clinical outcomes of COVID-19 using data from a global data sharing repository. Methods: We used data of all cases recorded in the Global Health Data repository up to the 10th of March 2021 to carry out a cross-sectional analysis of associations between cardiovascular diseases (CVD), hypertension, diabetes, obesity, lung diseases, and kidney disease and hospitalization, ICU admission, and mortality due to COVID-19. The Global Health repository reported data from 137 countries, but only Brazil, Mexico, and Cuba reported more than 10 COVID-19 cases in participants with preexisting conditions. We used multivariable logistic regression to compute adjusted odds ratios (aOR) of the three outcomes for each pre-existing condition in ten-year age groups from 0 to 9 years and up to 110–120 years. Findings: As of March 10, the Global Health repository contained 25,900,000 records of confirmed cases of COVID-19, of which 2,900,000 cases from Brazil, Mexico, and Cuba had recorded data on pre-existing conditions. The overall aOR of ICU admission for each pre-existing condition were; CVD (aOR 2.1, 95%CI 1.8–2.4), hypertension (aOR 1.3, 95%CI 1.2–1.4), diabetes (OR 1.7, 95%CI 1.5–1.8), obesity (OR 2.2, 95%%CI 2.1–2.4), kidney disease (OR 1.4, 95%CI 1.2–1.7) and lung disease (aOR 1.1, 95%CI 0.9–1.3). Overall aORs of mortality for each pre-existing condition were: CVD (aOR 1.7, 95%CI 1.6–1.7), hypertension (aOR 1.3, 95%CI 1.3–1.4), diabetes (aOR 2.0, 95%CI 1.9–2.0), obesity (aOR 1.9, 95%CI 1.8–2.0), kidney disease (aOR 2.7, 95%CI 2.6–2.9), and lung disease (aOR 1.6, 95%CI 1.5–1.7). The odds of each adverse outcome were considerably larger in children and young adults with these preexisting conditions than for adults, especially for kidney disease, CVD, and diabetes. Conclusion: This analysis of a global health repository confirms associations between pre-existing diseases and clinical outcomes of COVID-19, and the odds of these outcomes were especially elevated in children and young adults with these preexisting conditions. This study shows that global data sharing can unlock answers to many epidemiological questions efficiently especially during the early stages of global health emergencies.

## 1. Background

The availability of efficacious vaccines [1,2,3,4] and mounting immunity [5] to COVID-19 have blunted the impact of COVID-19, although pockets of vulnerability remain [3]. On 5 May 2023, the World Health Organization (WHO) declared an end to the COVID-19 global health emergency; however, abundant evidence suggests that the disease remains an ongoing global health challenge [6]. Recurring outbreaks of COVID-19 in various countries and regions [7] continue to put pressure on scarce health resources in many countries, while vulnerable populations remain disproportionately affected by COVID-19-related morbidity and mortality. The pandemic represents one of the very few instances in the modern era when global health security was tested, and the response, particularly during the early stages, was demonstrably inadequate.

The worldwide response to COVID-19, especially in the early months of the public health emergency, revealed a fractured and uncoordinated global health security system, which further fragmented into nationalistic and, at times, opposing responses at a time when unity was required [8,9]. A major weakness was the limited sharing of data across countries, which undermined the global health system by delaying the early detection of new pathogenic variants, rapid response, identification of transmission patterns, and accurate characterization of morbidity and mortality patterns [10]. Open data sharing during a public health emergency, such as COVID-19, could have yielded more timely answers to critical questions where evidence remained absent or uncertain, due to the unavailability of data, or where there was uncertain evidence from studies with small sample sizes [11]. Given the many advances in cloud computing and data analysis, sharing individual-level anonymized data in real time is not only feasible but also imperative in the context of a global health emergency of such magnitude. Data sharing could be made easy by collecting only a minimal set of data items per individual in a global data repository, provided that careful planning underpins such efforts. However, the degree to which such data repositories can effectively address epidemiological questions remains unclear.

To examine the utility of global health sharing, we used one international repository to answer an epidemiological question about the association between pre-existing chronic diseases and adverse outcomes of COVID-19. The association between pre-existing conditions and prognosis of COVID-19 is of considerable clinical and public health interest, as this could facilitate the rapid identification of at-risk patients at presentation. Ideally, such identification could be achieved through the integration of high-risk comorbidities and multimorbidity [12] with objective vital functional measures such as arterial blood gases, thereby improving prognostic discrimination at presentation [13,14,15]. Early identification of at-risk patients enables timely intervention and the more efficient allocation of scarce resources, such as oxygen and intensive care services (ICUs), during global health emergencies. Although a wealth of literature is available on the association between preexisting disease conditions and poor outcomes from COVID-19 infection, several gaps in knowledge still need to be explored. While age is generally accepted to be the most substantial risk factor for poor outcomes [16,17,18], evidence on the effect of preexisting conditions on adverse outcomes of COVID-19 in different age groups is limited. The risk of severe outcomes in children and young adults with comorbidities is also not well explored in current research. Notably, most available primary studies are limited by small sample sizes, with only a few exceptions [19,20,21], and data from low- and middle-income countries (LMICs) are underrepresented. In this study, we used an international data repository of COVID-19 cases from LMICs to investigate the association between preexisting conditions and three outcomes of COVID-19: hospitalization, need for ICU, and mortality.

## 2. Methods

### 2.1. Study Design and Setting

In this cross-sectional study, we used data from cases collected in the global health (GH) data repository [22,23]. The GH is a collaborative network of volunteers and organizations that aimed to enable real-time sharing of verifiable data on COVID-19 [23]. The full details of the methodology of the GH are described elsewhere [22]. In brief, the GH database is an open-source repository that was created to provide reliable, real-time data of verifiable and anonymized individual-level data on confirmed cases of COVID-19 [22]. Data were collected using official government reports, such as press releases, official websites, and official social media accounts, as well as primary data from peer-reviewed publications. Additional individual-level data were also collected from other sources, such as news websites, news aggregators, and peer-reviewed publications [22], and these were used to update case records. In the GH, data were captured by curators who are skilled in English, Mandarin Chinese, Cantonese, Spanish, and Portuguese, and both machine learning and manual checking were used to reduce duplicate entries and other errors in data capturing [22].

Although a standard data collection template was used in the GH repository, data from only a few countries included preexisting conditions. In this analysis, we included all cases in the GH database collected up to 10 March 2021 from countries where preexisting conditions and if the outcomes of interest were reported. We excluded cases for which data on country of residence, gender, or age were missing. We also excluded cases with age reported only in very broad categories, for example, 0–35 years.

### 2.2. Data Extraction and Data Management

The GH collected data on geographical location, demographics, dates of confirmation of COVID-19, travel history, preexisting conditions, and outcomes for each confirmed case. For each case, we extracted data on the following: age and gender, country of residence, preexisting conditions including diabetes, cardiovascular disease, hypertension, asthma, lung disease, and chronic kidney disease, and the three outcomes of interest, which were hospitalization, ICU admission, and mortality. A *for-loop* was used to extract data on each of the individual preexisting conditions into separate columns from the column of preexisting conditions in the GH dataset. Data on hospitalization and ICU admission were extracted from the relevant columns, and data on death were extracted from the “Outcome” column. For each case, additional data on preexisting conditions and outcomes were also extracted from the “Notes” variable using a *for-loop*. In the *for-loop*, we used text words and synonyms for each of the preexisting conditions and outcomes in languages used by the GH curators (Appendix A. Data were checked using summaries and frequency tables.

### 2.3. Data Analysis

We described categorical data using frequencies and percentages and compared groups using chi-square tests. Within each ten-year age group, we computed unadjusted and adjusted odds ratios (OR) and their 95% confidence intervals (95%CI) for each preexisting condition against the three binary outcomes (hospitalization, ICU admission, and mortality) using multivariable logistic regression. We used directed acyclic graphs to identify the minimum adjustment set for each association of interest. We adjusted for age, gender, and other preexisting conditions if they were confounders of the association under consideration. For the association between cardiovascular disease (CVD) and mortality, we adjusted for age, gender, obesity, hypertension, and diabetes. For the association between diabetes and mortality, we adjusted for gender, country, CVD, and obesity. For the association between lung diseases and mortality, we adjusted for age, gender, country, and obesity. For the association between kidney diseases and mortality, we adjusted for age, gender, country, CVD, diabetes, and obesity. For the association between obesity and mortality, we adjusted for age, gender, country, CVD, and diabetes. For the association between hypertension and mortality, we adjusted for age, gender, country, diabetes, and obesity. The complete Stata do-file for the analysis is in Appendix A.

### 2.4. Ethics

The ethical principles of the Declaration of Helsinki guided the conduct of this study [24]. Ethics approval and informed consent were not required as the study used deidentified data from a publicly available repository.

## 3. Results

A total of 25,774,885 case records from 137 countries were included in the GH repository as of 10 March 2021 (Figure 1). Most cases were diagnosed in 2020, with only 24% from the period January to March 2021. The demographic characteristics of these cases, stratified by hospitalization, ICU, and mortality, are shown in Table 1. Approximately half (47.3%) of the cases were female; 43.3% were male; and 9.2% had missing data on sex. Males, compared to females, had higher proportions of hospitalizations (3.2% vs. 2.8%), ICU admissions (0.3% vs. 0.2%), and mortality (2.1% vs. 1.6%). Compared with 2021, cases from 2020 had higher proportions of hospitalizations (2.9% vs. 2.2%), ICU admissions (0.2% vs. 0.1%), and mortality (1.9% vs. 1.1%). The countries contributing the largest proportion of cases to the repository were the USA (58.1%), Germany (9.5%), Colombia (8.6%), and Brazil (7.3%).

### 3.1. Characteristics of Cases from Countries Where Preexisting Conditions Were Reported

Ten countries had records of pre-existing conditions, while the remainder did not have such data (Figure 1 and Appendix A). Of these ten countries, only three countries, with a total of 1,919,628 cases, provided data on a substantial number of cases with pre-existing conditions: Mexico (N = 893,167), Brazil (N = 1,015,975), and Cuba (N = 10,486) (Appendix A); therefore, only cases from these countries were included in the analysis. Out of these cases, 338 cases had missing data on gender and were not included in the analysis, resulting in a total of 1,919,290. Just over half of the cases from the countries with pre-existing conditions were from Brazil (52.9%). The characteristics and pre-existing conditions of cases from these three countries are compared by outcome in Appendix A. Overall, 52.6% were female and 47.4% were male. Compared to females, males had higher proportions of hospitalization (5.1% vs. 3.6%, respectively), ICU admission (0.3% vs. 0.2%, respectively), and mortality (1.8% vs. 1.1%, respectively). Most of the cases were aged 30–39 years (22.9%), followed by those aged 40–49 years (19.8%), and 20–29 years (19.4%). The percentage of cases with severe outcomes (hospitalization, ICU admission, and mortality) increased with increasing age group (Appendix A).

### 3.2. Comparison of Hospitalisations, ICU, and Mortality by Pre-Existing Condition Status, Overall Analysis, and by Age Group

Overall, for all pre-existing conditions, there were higher proportions of cases with hospitalizations, intensive care services, and mortality, compared to cases without pre-existing conditions (Appendix A). For example, cases with CVD, compared to cases without CVD, were more likely to be hospitalized (17.5% vs. 3.0%, respectively, *p* < 0.01), to need ICU (0.9% vs. 0.2%, respectively, *p* < 0.01), and to die from COVID-19 (6.9% vs. 0.9%, respectively, *p* < 0.01). Cases with diabetes, compared to cases without diabetes, were more likely to be hospitalized (21.4% vs. 3.3%, respectively, *p* < 0.01), more likely to need ICU (1.1% vs. 0.2%, respectively, *p* < 0.01), and had higher mortality from COVID-19 (7.6% vs. 1.0%, respectively, *p* < 0.01). This pattern was observed across all pre-existing conditions, and more so in cases with kidney disease, compared to cases without kidney disease, appeared to have very higher vulnerability, with high proportions hospitalized (44.7% vs. 4.1%, respectively, *p* < 0.01), in ICU (1.5% vs. 0.2%, respectively, *p* < 0.01) and high mortality (15.6% vs. 1.3%, respectively, *p* < 0.01) (Appendix A).

The proportions of individuals with adverse outcomes (hospitalization, ICU, and mortality) were especially high in younger age groups, including children and young adults with pre-existing diseases, relative to those without pre-existing diseases (Appendix A).

### 3.3. Association Between Pre-Existing Conditions and Hospitalization, ICU Admission, and Mortality from COVID-19

The overall adjusted odds of hospitalization for each pre-existing condition were as follows: CVD (aOR 1.7, 95%CI 1.7–1.7), hypertension (aOR 1.5, 95%CI 1.4–1.5), diabetes (aOR 2.2, 95%CI 2.1–2.2), obesity (aOR 1.7, 95%CI 1.6–1.7), kidney disease (aOR 5.5, 95%CI 5.2–5.7) and lung disease (aOR 1.9, 95%CI 1.8–1.9) (Figure 2) (Table 2 and Appendix A). Similarly, the overall adjusted odds of ICU admission for each pre-existing condition were as follows: CVD (aOR 2.1, 95%CI 1.8–2.4), hypertension (aOR 1.3, 95%CI 1.2–1.4), diabetes (aOR 1.7, 95%CI 1.5–1.8), obesity (aOR 2.2, 95%CI 2.1–2.4), kidney disease (aOR 1.4, 95%CI 1.2–1.7) and lung disease (aOR 1.1, 95%CI 0.9–1.3) (Figure 3 and Appendix A). Finally, the overall adjusted odds of mortality for each pre-existing condition were as follows: CVD (aOR 1.7, 95%CI 1.6–1.7), hypertension (aOR 1.3, 95%CI 1.3–1.4), diabetes (aOR 2.0, 95%CI 1.9–2.0), obesity (aOR 1.9, 95%CI 1.8–2.0), kidney disease (aOR 2.7, 95%CI 2.6–2.9), and lung disease (aOR 1.6, 95%CI 1.5–1.7) (Figure 3 and Appendix A). The odds of each outcome were considerably higher in younger age groups, including children and young adults with these preexisting conditions, compared to adults, especially for kidney disease, CVD, and diabetes (Figure 3 and Appendix A).

## 4. Discussion

In this cross-sectional analysis, we demonstrate the utility of a global health data sharing repository for COVID-19 cases in answering key epidemiological questions. The analysis confirmed that individuals with pre-existing CVD, diabetes, obesity, hypertension, kidney disease, and lung diseases have higher odds of hospitalization, ICU admission, and death from COVID-19 outcomes compared to those with no pre-existing comorbidities.

Kidney diseases were associated with the highest odds of mortality from COVID-19, followed by diabetes, obesity, CVD, and lung diseases. Our findings are consistent with several meta-analyses, although some of these meta-analyses have produced inconclusive results due to limited available studies, most of which included small samples [25,26,27,28,29,30,31,32,33]. What is clear and demonstrated by the current study and other similar studies is that most people who experience worse COVID-19 outcomes have at least one pre-existing condition [34]. A drawback of meta-analyses is that they rely on sufficient accumulation of primary research studies [35], and therefore, they are not a ready source for answers to clinical and public health questions during the early phases of a pandemic. Further, the quality of evidence from meta-analyses depends largely on the quality of included primary studies [36], and if those primary studies are of poor quality, as was largely true during the early stages of the COVID-19 pandemic [37,38], then the meta-analysis could be misleading. A global data sharing repository is therefore a superior resource and likely to provide better quality evidence if thoughtfully implemented.

We found that children and young adults with pre-existing conditions had higher odds of severe COVID-19 outcomes. For example, children and young adults with kidney diseases had almost a 20-fold increase in the odds of COVID-19 mortality compared to those without these comorbidities. Data on children with comorbidities have been more limited than in adults, perhaps due to the relatively low mortality from COVID-19 in children, and therefore, single-centre studies would not have had the power to detect the effects of pre-existing conditions on such an outcome [39,40,41]. The global health repository offered an obvious advantage in this scenario, as we were able to leverage the power of large case counts from different countries to estimate the effect of pre-existing conditions on mortality and other adverse outcomes in children. Our results show that in children with pre-existing comorbidities such as renal failure, heart diseases, and diabetes, the effect of COVID-19 is worse than in adults, despite the protective effect of their young age. These are not novel findings, as this has been documented in some prior research. Some studies have shown that pre-existing chronic diseases superseded age in their effect on in-hospital COVID-19 mortality [42,43].

There are several possible reasons why children with comorbidities have a poor prognosis when infected with COVID-19, compared to adults with the same comorbidities. Children, in general, tend to have immature immune systems, with full immune competency being attained well into adolescence [44]. Pre-existing conditions compound this by reducing the children’s immune system’s ability to produce an effective response to viral infections such as COVID-19 and influenza, as reported in several studies [45,46]. Some pre-existing conditions increase the risk of hyperinflammatory reactions that only happen in children, such as multisystem inflammatory syndrome in children [47], which likely contributes to the worsened COVID-19 prognosis in children. Further, children with pre-existing conditions are likely to have reduced physiological reserve, for example, reduced baseline lung or cardiac capacity, and viral infections resulting in mild respiratory distress could rapidly progress to organ failure in these children, as seen in sepsis [48]. In addition, children with pre-existing conditions are likely to have lifelong, severe, genetic diseases such as congenital disorders, which may have impaired their physical and immunological development, and this may worsen COVID-19 prognosis [49]. In adults, many pre-existing conditions such as type 2 diabetes, heart disease, and renal disease may have developed later, after many healthy years, which may partly explain their relatively better COVID-19 prognosis compared to children with lifelong conditions. Other reasons why children with pre-existing diseases may have worse outcomes include therapeutic limitations and uncertainty in children during the early months of the COVID-19 pandemic, and difficulties in recognizing or distinguishing between underlying disease exacerbation and COVID-19 disease worsening [50,51].

By the end of the first quarter of 2020, the GH repository contained almost half a million cases of COVID-19 (473,345 cases). At this time, based on published reports from single-centre studies, many of the associations investigated in the current study did not have conclusive findings, and many conclusions were drawn from small samples with uncertain results. Before the current analysis, the debate on some of these associations remained inconclusive for some of the pre-existing conditions. Our findings suggest that a lot of research waste and clinical uncertainty would have been avoided had these data been well used during the early stages of the COVID-19 pandemic. For example, children were generally assumed to be safe from COVID-19, but our analysis clearly shows that children with comorbidities should have received greater protection. Another key use of findings from such a data-sharing technology is in the rapid identification of the at-risk patient through the incorporation of high-risk pre-existing conditions data with vital data at presentation, such as arterial blood gases [15]. This is a critical need, given the scarcity of health resources in a global health emergency. This study lends further credence to the need for data sharing and the need for leveraging data technologies to ensure better and faster sharing of data to inform clinical and policy decision-making, as well as mitigatory measures in health emergencies.

Some strengths of this study include the large sample size and the robust analysis that we carried out. In this study, we have demonstrated that this type of data sharing, if implemented early during a global health emergency such as COVID-19, could facilitate quicker and more precise answers not only to questions about disease etiology, transmission, morbidity, and mortality, but also to many epidemiological questions such as the effect of pre-existing chronic diseases, as illustrated in the current study. However, some limitations of these data repositories include the difficulties in verifying case data. Undiagnosed pre-existing conditions may not have been captured, and a lack of uniformity in data collection may render some data unusable, as observed in the case of data from the USA, Germany, and other countries where pre-existing diseases were not reported. Further, data on vaccination status were not available, primarily because most of the study cohort was from 2020, before widespread vaccination. Additional limitations include a lack of data on disease severity, medications, and long-term follow-up. This is likely due to constraints in collecting data using one form from different health systems during a global health emergency. Assembling the smallest possible dataset, i.e., the fewest columns, while capturing the most useful data would have been the priority at that time, compared with creating a comprehensive dataset. Future work could include longitudinal follow-up of cases to study the long-term effects of COVID-19, as well as provide data on risk factors and prevalence of long COVID-19. Ecological analysis could also provide insight into how population-level COVID-19 outcomes are influenced by the type and organization of healthcare systems, the proportion of the gross domestic product spent on healthcare, and country-level vaccination rates.

## 5. Conclusions

This analysis of a global health repository confirms associations between pre-existing conditions and clinical outcomes of COVID-19. The odds of these outcomes were especially elevated in children and young adults with these preexisting conditions. In times of health emergencies, global data sharing may provide quicker and more precise estimates of epidemiological parameters. Global data sharing could be enhanced through the establishment of common legal and ethical frameworks for data sharing [52], as well as the use of standardized data collection systems and interoperable data systems [53]. Other ways of enabling global data sharing include creating trusted, neutral data repositories [54], and increasing investment in the digital infrastructure needed for data collection, management, storage, and protection, as well as human resources.

## Figures and Tables

**Figure 1 pathogens-14-00917-f001:**
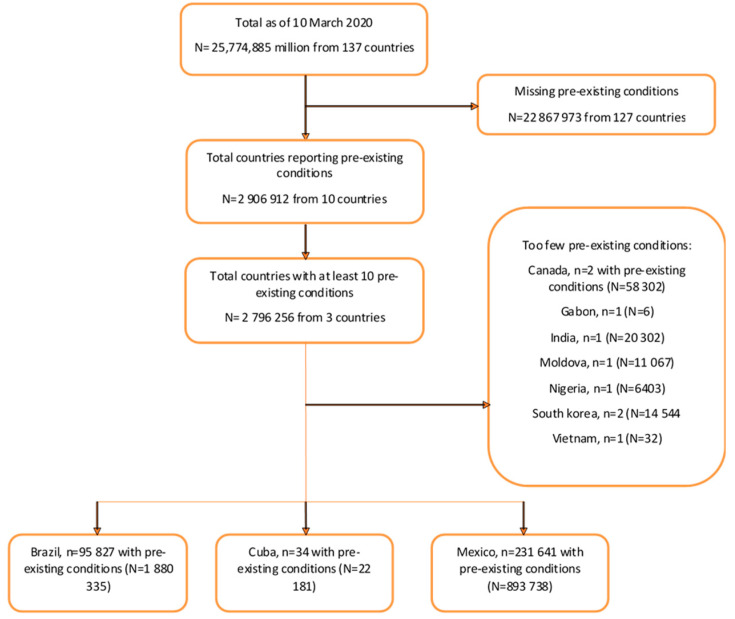
Flow chart: characteristics of cases in the whole global health repository.

**Figure 2 pathogens-14-00917-f002:**
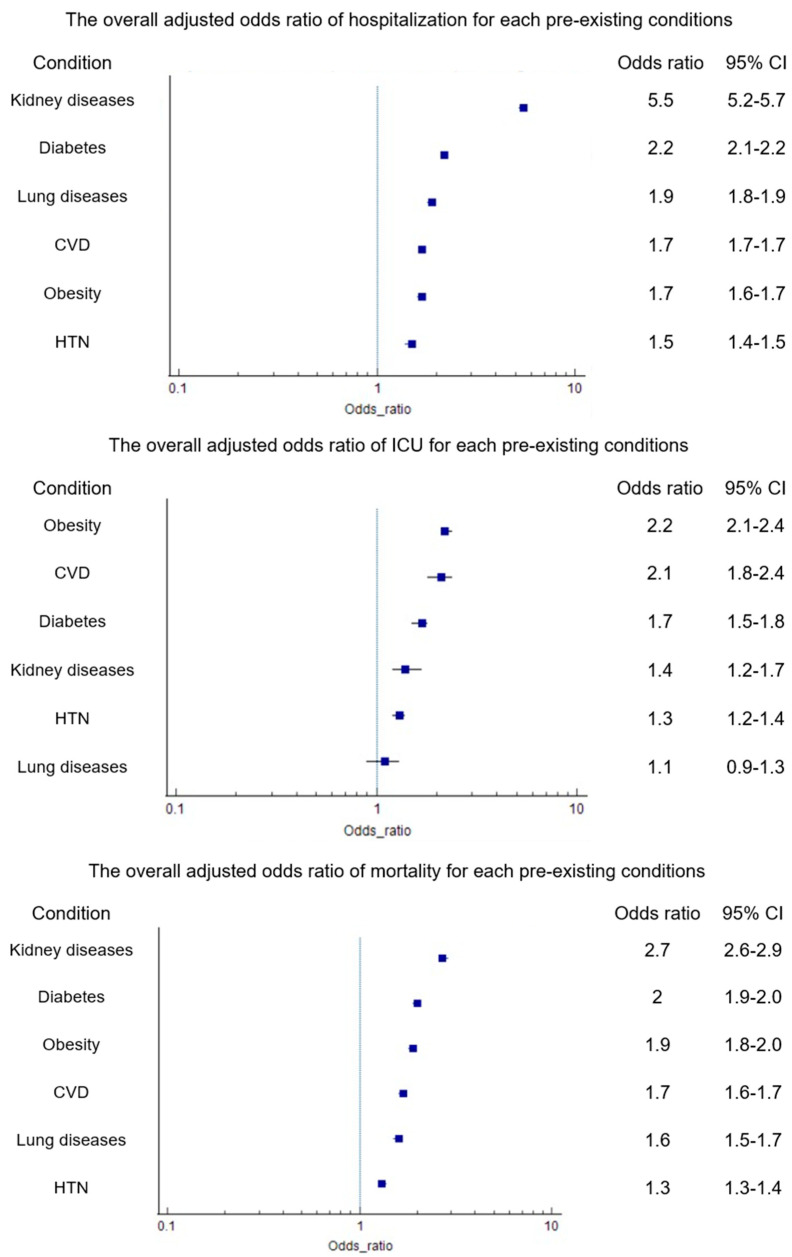
The overall adjusted odds ratios of hospitalization, ICU, and mortality for each pre-existing condition. CVD adjusted for gender, country, diabetes, and obesity. Lung diseases adjusted for gender, country, and obesity. Diabetes adjusted for gender, country, CVD, and obesity. Kidney diseases adjusted for gender, country, CVD, diabetes, and obesity. Obesity adjusted for gender, country, CVD, and diabetes. HTN adjusted for gender, country, diabetes, and obesity. Abbreviations: CVD (cardiovascular disease, including hypertension), HTN (hypertension).

**Figure 3 pathogens-14-00917-f003:**
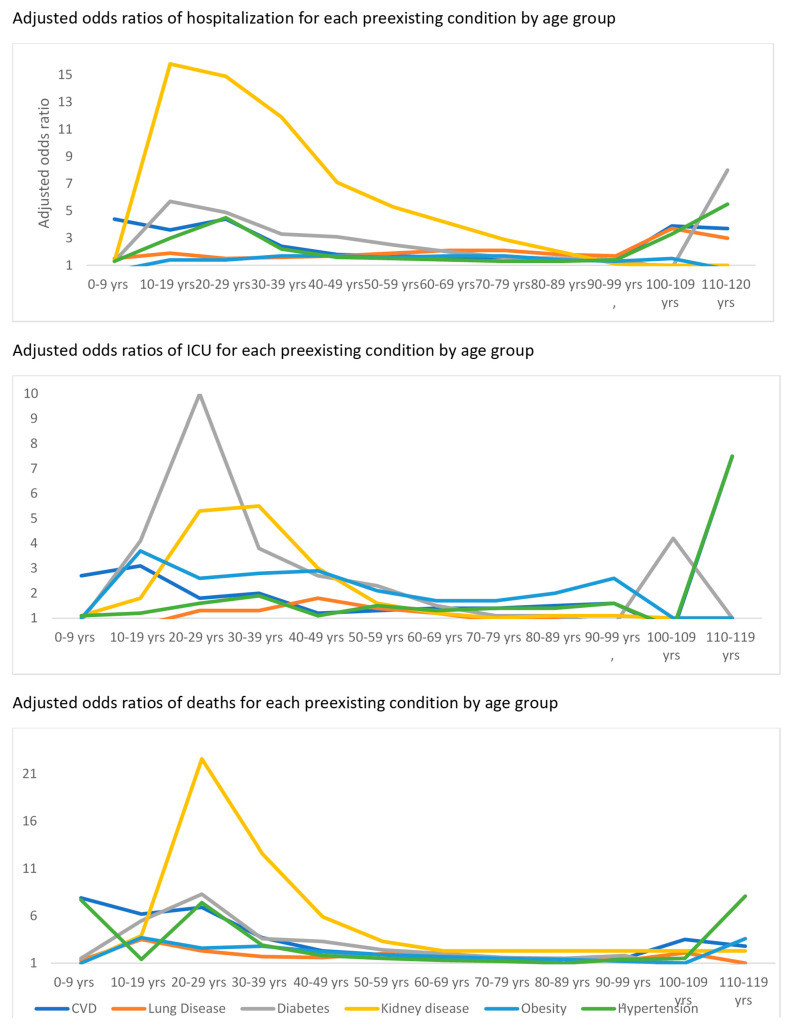
Adjusted odds ratios of hospitalization, ICU, and mortality for each pre-existing condition by age group. CVD adjusted for gender, country, diabetes, and obesity. Lung diseases adjusted for gender, country, and obesity. Diabetes adjusted for gender, country, CVD, and obesity. Kidney diseases adjusted for gender, country, CVD, diabetes, and obesity. Obesity adjusted for gender, country, CVD, and diabetes. HTN adjusted for gender, country, diabetes, and obesity. Abbreviations: CVD (cardiovascular disease, including hypertension).

**Table 1 pathogens-14-00917-t001:** Characteristics of cases in the global health dataset as of 10 March 2021.

	Hospitalization		ICU		Mortality		Total, N = 25,774,885
	Yes, N = 692,001 (2.7%)	No, N = 25,082,884 (97.3%)	Yes, N = 50,195 (0.2%)	No, N = 25,724,690 (99.8%)	Yes, N = 433,394 (1.7%)	No, N = 25,341,491 (98.3%)	
Gender, Female, n (%)	335,165 (2.8)	11,853,326 (97.3)	20,646 (0.2)	12,167,845 (99.8)	192,446 (1.6)	11,996,045 (98.4)	12,188,491 (47.3)
Gender, Male, n (%)	354,059 (3.2)	10,870,336 (96.9)	29,433 (0.3)	11,194,962 (99.7)	240,130 (2.1)	10,984,265 (97.9)	11,224,395 (43.5)
Gender, Others, n (%)	13 (4.3)	287 (95.7)	4 (1.3)	296 (98.7)	5 (1.7)	295 (98.3)	300 (0.0)
Gender, Missing, n (%)	2764 (0.1)	2,358,935 (99.9)	112 (0.0)	2,361,587 (100.0)	813 (0.0)	2,360,886 (100.0)	2,361,699 (9.2)
Year of diagnosis							
2020, n (%)	557,643 (2.9)	19,021,218 (97.2)	43,021 (0.2)	19,535,840 (99.8)	366,883 (1.9)	19,211,978 (98.1)	19,578,861 (76.0)
2021, n (%)	134,358 (2.2)	6,061,666 (97.8)	7174 (0.1)	6,188,850 (99.9)	66,511 (1.1)	6,129,513 (98.9)	6,196,024 (24.0)
Period of diagnosis							
Jan–Mar 2020	27,372 (5.8)	445,973 (94.2)	3264 (0.7)	470,081 (99.3)	11,807 (2.5)	461,538 (97.5)	473,345 (1.8)
Apr–Jun 2020	167,268 (4.7)	3,399,956 (95.3)	17,993 (0.5)	3,549,231 (99.5)	109,771 (3.1)	3,457,453 (96.9)	3,567,224 (13.8)
Jul–Sep 2020	122,355 (2.4)	4,907,294 (97.6)	9271 (0.2)	5,020,378 (99.8)	92,153 (1.8)	4,937,496 (98.2)	5,029,649 (19.5)
Oct–Dec 2020	240,648 (2.3)	10,267,995 (97.7)	12,493 (0.1)	10,496,150 (99.9)	153,152 (1.5)	10,355,491 (98.5)	10,508,643 (40.8)
Jan–Mar 2021	134,358 (2.2)	6,061,666 (97.8)	7174 (0.1)	6,188,850 (99.9)	66,511 (1.1)	6,129,513 (98.9)	6,196,024 (24.0)
Most considerable case contributions by country							
USA, n (%)	587,195 (3.9)	14,388,175 (96.1)	43,152 (0.3)	14,932,218 (99.7)	259,753 (1.7)	14,715,617 (98.3)	14,975,370 (58.1)
Germany, n (%)	0	2,448,424 (100.0)	0	2,448,424 (100.0)	65,191 (2.7)	2,383,233 (97.3)	2,448,424 (9.5)
Colombia, n (%)	17,883 (0.8)	2,211,189 (99.2)	2770 (0.1)	2,226,302 (99.9)	66,943 (3.0)	2,162,129 (97.0)	2,229,072 (8.6)
Brazil, n (%)	6286 (0.3)	1,874,049 (99.7)	61 (0.0)	1,880,274 (100.0)	9016 (0.5)	1,871,319 (99.5)	1,880,335 (7.3)

NB: *p*-values omitted because all *p*-values were highly significant due to the big sample sizes in each comparison.

**Table 2 pathogens-14-00917-t002:** Overall adjusted odds ratios of hospitalization, ICU admission, and mortality for each pre-existing condition.

Overall	Hospitalization	ICU Admission	Mortality
	aOR (95% CI)	aOR (95% CI)	aOR (95% CI)
Cardiovascular diseases	1.7 (1.7–1.7)	1.4 (1.3–1.5)	1.7 (1.6–1.7)
Lung diseases	1.9 (1.8–1.9)	1.1 (0.9–1.3)	1.6 (1.5–1.7)
Diabetes	2.2 (2.1–2.2)	1.7 (1.5–1.8)	2.0 (1.9–2.0)
Kidney diseases	5.5 (5.2–5.7)	1.4 (1.2–1.7)	2.7 (2.6–2.9)
Obesity	1.7 (1.6–1.7)	2.2 (2.1–2.4)	1.9 (1.8–2.0)
Hypertension	1.5 (1.4–1.5)	1.3 (1.2–1.4)	1.3 (1.3–1.4)

## Data Availability

All data used in this analysis are available from the global health repository—https://data.covid-19.global.health/cases, accessed on 10 March 2021.

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
