# Peer review of "Association Between Pre-Existing Conditions and COVID-19 Hospitalization, Intensive Care Services, and Mortality: A Cross-Sectional Analysis of an International Global Health Data Repository"

_pathogens, 2025, doi:10.3390/pathogens14090917_

Round 1

Reviewer 1 Report

Comments and Suggestions for Authors

Pathogen-3816492:
Association between pre-existing conditions and COVID-19 hospitalization, intensive care, and mortality: A cross-sectional analysis of the International Global Health Data Repository (Basant M. S. Elsayed et al.)

Overall comments:
The authors investigated the association between pre-existing conditions and clinical outcomes of COVID-19. In this cross-sectional analysis, the authors demonstrated the usefulness of international health data repositories for answering key epidemiological questions. The analysis confirmed that individuals with pre-existing conditions, such as CVD, diabetes, obesity, hypertension, kidney disease, and lung disease, had higher odds of COVID-19 hospitalization, ICU admission, and death compared with individuals without these conditions. While this study provides interesting insights for global health research, several issues and questions need to be clarified.

Specific comments:
1)    Information on vaccination status (time of vaccination and time since last dose) and acute treatment for infection should be included.

2)    Information on the severity and main symptoms of COVID-19 would be useful in discussing the current database. Information on medications taken before infection (number of doses and the breakdown) is also necessary to discuss this issue.

3)    It would be interesting to discuss why pre-existing conditions differ by age in this analysis. How did the activity and treatment status of each pre-existing condition change?

4)    Gender differences in pre-existing conditions should also be analyzed to clarify the underlying factors behind COVID-19 exacerbation.

5)    To verify the transition from COVID-19 to long-term COVID, it is necessary to track changes in symptoms after infection.

6)    The statistical analysis of the data presented in the figure appears insufficient. Regardless of whether the presented results were statistically significant, the authors should clearly demonstrate the comparison of the data.

7)    To facilitate reader understanding, we recommend that reviewers provide a figure outlining the data and its significance to summarize and visualize the presented data.

Author Response

Reviewer 1

Overall comments:

The authors investigated the association between pre-existing conditions and clinical outcomes of COVID-19. In this cross-sectional analysis, the authors demonstrated the usefulness of international health data repositories for answering key epidemiological questions. The analysis confirmed that individuals with pre-existing conditions, such as CVD, diabetes, obesity, hypertension, kidney disease, and lung disease, had higher odds of COVID-19 hospitalization, ICU admission, and death compared with individuals without these conditions. While this study provides interesting insights for global health research, several issues and questions need to be clarified.

Specific comments:

  • Information on vaccination status (time of vaccination and time since last dose) and acute treatment for infection should be included.

Thank you. While we agree that this information would have been greatly beneficial, the data were not available in the dataset. This is most likely because most of the individuals in our analysis dataset had COVID-19 during 2020, before widespread vaccination campaigns began. We have included this as a limitation, as shown below:

Further, data on vaccination status were not available, mainly due to most of the study cohort being from 2020, before widespread vaccination.

2)    Information on the severity and main symptoms of COVID-19 would be useful in discussing the current database. Information on medications taken before infection (number of doses and the breakdown) is also necessary to discuss this issue.

Thank you. While we agree that this information would have been greatly beneficial, again, similar to the first point above, the data were not available in the dataset. This is most likely because of limitations in collecting data using one form from different health systems during a global health emergency. Assembling the smallest possible dataset, i.e. the fewest columns in such a way that the most useful data could be captured would have been the bigger priority at that time, compared to a comprehensive dataset. We have included this as a limitation, as shown below:

Further limitations include a lack of data on severity, medications and long-term follow-up. This is most likely because of limitations in collecting data using one form from different health systems during a global health emergency. Assembling the smallest possible dataset, i.e. the fewest columns, in such a way that the most useful data could be captured would have been the bigger priority at that time, compared to a comprehensive dataset.

3)    It would be interesting to discuss why pre-existing conditions differ by age in this analysis. How did the activity and treatment status of each pre-existing condition change?

Thank you. We have added to the discussion some possible reasons why children with pre-existing conditions may have worse prognosis, see below:

There are several possible reasons why children with comorbidities had higher odds of adverse clinical outcomes, such as mortality, compared to adults, when infected with COVID-19. Children in general tend to have immature immune systems, with full immune competency being attained well into adolescence (34). Pre-existing conditions compound this by reducing the children’s immune system’s ability to produce an effective response to viral infections such as COVID-19 and influenza, as reported in several studies (35, 36). Some pre-existing conditions increase the risk of hyperinflammatory reactions that only happen in children, such as multisystem inflammatory syn-drome in children (37), which likely contributed to the worsened COVID-19 prognosis in children. Children with pre-existing conditions are likely to have reduced physiological reserve, for example, reduced baseline lung or cardiac capacity, and viral infections resulting in mild respiratory distress could rapidly cause organ failure in these children, as seen in sepsis (38). Further, children with pre-existing conditions are likely to have lifelong, severe and genetic diseases such as congenital disorders, which would have impaired their physical and immunological development, and again likely worsen COVID-19 prognosis (39). In adults, many pre-existing conditions such as type 2 diabetes, heart disease, and renal disease may have developed later, after some healthy years. Other reasons why children with pre-existing diseases may have worse outcomes include therapeutic limitations and uncertainty in children during the early months of the COVID-19 pandemic, and difficulties in recognizing or distinguishing between under-lying disease exacerbation and COVID-19 disease worsening (40) (41)     

Similar to the above comment in (2), we agree that an analysis of treatment status would have been invaluable. However, please note that this study was not designed to provide causal associations but to test the feasibility of testing causal associations given a globally shared dataset with all its limitations, including the lack of granular data such as treatment status. This is discussed as a limitation, as shown below:

Further limitations include a lack of data on severity, medications and long-term follow-up. This is most likely because of limitations in collecting data using one form from different health systems during a global health emergency. Assembling the smallest possible dataset, i.e. the fewest columns, in such a way that the most useful data could be captured would have been the bigger priority at that time, compared to a comprehensive dataset.

4)    Gender differences in pre-existing conditions should also be analyzed to clarify the underlying factors behind COVID-19 exacerbation.

Thank you. Please note that we adjusted for gender in every association. More details of the adjustments have been added to the tables and figures where the multivariable regression results are reported.

5)    To verify the transition from COVID-19 to long-term COVID, it is necessary to track changes in symptoms after infection.

Thank you. We agree and have added this to the discussion as a suggestion for future work.

6)    The statistical analysis of the data presented in the figure appears insufficient. Regardless of whether the presented results were statistically significant, the authors should clearly demonstrate the comparison of the data.

Thank you. We have carried out detailed statistical analyses, and these analyses are shown in the supplementary materials. We have added more details to the methods in the main paper, to improve clarity and help address this point. 

7)    To facilitate reader understanding, we recommend that reviewers provide a figure outlining the data and its significance to summarize and visualize the presented data.

We thank the reviewer for this and kindly suggest that this is not needed, as we have submitted several figures already for this purpose (please see Figs 1, 2, 3). Within figures 2 and 3, there are multiple figures showing each of the analyses that we carried out. We have added legends to explain each of these figures in detail

Reviewer 2 Report

Comments and Suggestions for Authors

A possibly interesting study...I do have some comments to make on it:

1) Were you able to compare statistically the countries themselves with each other? With adjusted statistics? You did with Brazil, Cuba, and Mexico...what about the other countries?

2) Did you take into account the different healthcare systems from each of the countries? The # of GDP spent on healthcare? The costs associated with treating COVID-19 and other comorbidities? The costs associated with vaccination? Population sizes?

3) Were overall vaccination rates taken account for each country?

4) You conclude with the need for better data sharing between countries...You should provide a couple of examples of how countries can do this.

Author Response

  • Were you able to compare statistically the countries themselves with each other? With adjusted statistics? You did with Brazil, Cuba, and Mexico...what about the other countries?

Thank you. We have clarified that these three countries were the ones which provided data on pre-existing conditions. The other countries did not have data on pre-existing conditions in the dataset, so they were not included in the analysis of the effect of pre-existing conditions on clinical outcomes.

  • Did you take into account the different healthcare systems from each of the countries? The # of GDP spent on healthcare? The costs associated with treating COVID-19 and other comorbidities? The costs associated with vaccination? Population sizes?

Thank you, this has been included in the discussion as it was not part of the analysis of this study. Please note that this is not an ecological comparison of the outcomes per country, where such suggested variables can then be adjusted for. We do agree that such an analysis would be valuable; however, that is beyond the scope of the current study, whose main focus was testing the feasibility of using rapidly collected and shared global data to answer epidemiological questions.

3) Were overall vaccination rates taken account for each country?

Thank you. We did not take vaccination into account, as most of our data were from pre-vaccination periods of the COVID-19 pandemic. Individual vaccination status was not available in the dataset at the time the dataset was compiled. Again, an ecological analysis of country-level vaccination rates versus country-level clinical outcomes is possible and answers a different question from the one we asked in this paper. The ecological analysis is not the scope of this study

  • You conclude with the need for better data sharing between countries...You should provide a couple of examples of how countries can do this.

Thank you. This has been added to the discussion as shown below:

Global data-sharing could be enhanced through the establishment of common legal and ethical frameworks for data-sharing (Vlahou et al., 2021), the use of data collection systems and interoperable data systems (Brenas et al., 2017), creating trusted, neutral data repositories (Pisani et al., 2018), and more investments in the digital infrastructure needed for data collection, management, storage and protection as well as human resources.  

Reviewer 3 Report

Comments and Suggestions for Authors

Authors performed an analysis of the GH data repository, to find any associations between comorbidities and ICU admission/death due to COVID-19. The study is not a novelty and there are some (also methodological) issues. Finally, language should be improved (i.e. there’re many repeated terms such as “global”

First of all, authors should check instruction for authors (https://www.mdpi.com/journal/pathogens/instructions), abstract should be less than 200 words , brackets should be [ ] and not (), check all the consecutive references listed in order they appear etc.

Introduction: authors should mentioned the importance of a rapid identification of patients at risk to develop a severe COVID-19 form, not only based on comorbidities, but also from blood gas analysis (BGA) yet in emergency room setting as  Alveolar-arterial gradient (10.3390/idr14030050) and PaO2/FiO2 (10.3390/biomedicines13051072).

Authors stated that “Additional data of individual cases were also collected from other sources such as news websites, news aggregators and peer reviewed publications and used to update data of cases.”  (row 98-99). Do you add “other patients” to your cohort? If yes, this could pose a methodological issues, so clarify how do you assessed the risk of “overlap” or “duplicate” cases and how it affected the final results.

Methods: do you performed a Yates correction for chi-square? 

Explain exactly what do you mean: do you exclude “patient without comorbidities” or data from repository where comorbidities weren’t reported?

How many patients without comorbidities do you find?

Finally, do you include patients “without comorbidities” ?

Results:

when comparing two groups (i.e. row 153-154-156 etc.) you might include statistical significance (p value).

Figure 1: “total countries with at least 10 pre-existing conditions”. It means that you inlcude only nation with 10 pre-existing conditions? 

Table 1: you should include the number of patients with each comorbidities (total, ICU, death, hospitalization) ; a p. value should be included on the right 

Discussion:

Discuss about the importance of a rapid identification of a “at-risk” patient (from anamnesis -comorbidities- and BGA - alveolar arterial gradient and P/F)

Author Response

Authors performed an analysis of the GH data repository, to find any associations between comorbidities and ICU admission/death due to COVID-19. The study is not a novelty and there are some (also methodological) issues. Finally, language should be improved (i.e. there’re many repeated terms such as “global”

Thank you. We have reviewed the manuscript for grammar and conciseness

First of all, authors should check instruction for authors (https://www.mdpi.com/journal/pathogens/instructions), abstract should be less than 200 words , brackets should be [ ] and not (), check all the consecutive references listed in order they appear etc.

 Thank you, we have revised the manuscript and ensured these guidelines are followed

Introduction: authors should mentioned the importance of a rapid identification of patients at risk to develop a severe COVID-19 form, not only based on comorbidities, but also from blood gas analysis (BGA) yet in emergency room setting as  Alveolar-arterial gradient (10.3390/idr14030050) and PaO2/FiO2 (10.3390/biomedicines13051072).

Thank you, we have added this to the introduction

 The association between pre-existing conditions and prognosis of COVID-19 is of strong interest, as this could help with the rapid identification of at-risk patients at presentation. The rapid identification could be done, ideally, through the integration of high-risk comorbidities and multimorbidity (AbouGalala et al., 2023) with objective vital functional measures such as arterial blood gases for improved prognostic discrimination at presentation (Liu et al., 2023). Early identification of at-risk patients enables early intervention and judicious use of scarce resources such as oxygen and intensive care services (ICU) in global health emergencies.

Authors stated that “Additional data of individual cases were also collected from other sources such as news websites, news aggregators and peer reviewed publications and used to update data of cases.”  (row 98-99). Do you add “other patients” to your cohort? If yes, this could pose a methodological issues, so clarify how do you assessed the risk of “overlap” or “duplicate” cases and how it affected the final results.

Thank you. No, we did not add patients to the global health cohort. However, this section describes how data on cases were collected by volunteers in the global health repository. We also added that “Data were captured by curators who are skilled in English, Mandarin Chinese, Cantonese, Spanish and Portuguese and machine learning and manual checking were used to reduce duplicate entries and other errors in data capturing (Xu et al., 2020).”

Methods: do you performed a Yates correction for chi-square?

Thank you. No, the Yates correction for the chi-square test was not needed because the study had large sample sizes, which made any continuity correction redundant

Explain exactly what do you mean: do you exclude “patient without comorbidities” or data from repository where comorbidities weren’t reported?

Thank you. We meant that we excluded data from repositories where comorbidities were not reported.

How many patients without comorbidities do you find?

Thank you. These data are presented in Supplementary Tables 3 -5, and as can be seen, vary depending on the comorbidity of interest and the association under consideration.

Finally, do you include patients “without comorbidities” ?

Thank you. Yes, to analyze the effect of comorbidities on COVID-19 clinical outcomes, the comparison was between patients with comorbidities compared to those without comorbidities, as shown in Supplementary Tables 3-5.

Results:

when comparing two groups (i.e. row 153-154-156 etc.) you might include statistical significance (p value).

Thank you, we opted, for brevity, to omit p-values because all p-values were highly significant, principally because of the huge sample sizes involved. We have added this under Table 1.

Figure 1: “total countries with at least 10 pre-existing conditions”. It means that you include only nations with 10 pre-existing conditions?

Thank you. Yes, in Table 2 and all further analyses, we included only nations where at least ten pre-existing conditions were reported, as that would allow us to analyze at least one comorbidity and adjust for some confounders using multivariable logistic regression

Table 1: you should include the number of patients with each comorbidities (total, ICU, death, hospitalization) ; a p. value should be included on the right

Thank you. We initially included these data in Table 1, but it became too big and could not fit into the paper. These data that the reviewer suggests are in Supplementary Tables 3-5

Discussion:

Discuss about the importance of a rapid identification of a “at-risk” patient (from anamnesis -comorbidities- and BGA - alveolar arterial gradient and P/F)

Thank you. We have added this to the discussion.

Another key use of findings from such a data-sharing technology is in the rapid identification of the at-risk patient through the incorporation of high-risk pre-existing conditions data with vital data at presentation such as arterial blood gases(Marra et al., 2025). This is a critical need given scarce health resources in a global health emergency.

Round 2

Reviewer 1 Report

Comments and Suggestions for Authors

The manuscript was appropriately revised based on the referees' opinions.

Author Response

Thank you

Reviewer 2 Report

Comments and Suggestions for Authors

Corrected all recommended suggestions....

Author Response

Thank you 

Reviewer 3 Report

Comments and Suggestions for Authors

Accepted

Author Response

Thank you